# Isolation of *Epizootic Hemorrhagic Disease Virus* Serotype 10 from *Culicoides tainanus* and Associated Infections in Livestock in Yunnan, China

**DOI:** 10.3390/v16020175

**Published:** 2024-01-24

**Authors:** Yuwen He, Jinxin Meng, Nan Li, Zhao Li, Dongmei Wang, Meiling Kou, Zhenxing Yang, Yunhui Li, Laxi Zhang, Jinglin Wang

**Affiliations:** 1Yunnan Tropical and Subtropical Animal Viral Disease Laboratory, Yunnan Animal Science and Veterinary Institute, Kunming 650224, China; heyuwen@foxmail.com (Y.H.); mengjinxin1989@163.com (J.M.); linan691@126.com (N.L.); ladyjanelili@163.com (M.K.); s300yn@163.com (Z.Y.); 2Jiangcheng County Animal Disease Prevention and Control Center, Jiangcheng 665900, China; lz995519@163.com (Z.L.); dongd202312@163.com (D.W.); lyh13769950404jj@163.com (Y.L.); znx13769927257@163.com (L.Z.)

**Keywords:** *epizootic hemorrhagic disease virus*, *Culicoides tainanus*, full genome analysis, infections in Livestock, virus isolation and identification

## Abstract

Two strains of viruses, JC13C644 and JC13C673, were isolated from *Culicoides tainanus* collected in Jiangcheng County, Yunnan Province, situated along the border area shared by China, Laos, and Vietnam. JC13C644 and JC13C673 viruses can cause cytopathic effect (CPE) in mammalian cells BHK21 and Vero cells, and cause morbidity and mortality in suckling mice 48 h after intracerebral inoculation. Whole-genome sequencing was performed, yielding complete sequences for all 10 segments from Seg-1 (3942nt) to Seg-10 (810nt). Phylogenetic analysis of the sub-core-shell (T2) showed that the JC13C644 and JC13C673 viruses clustered with the *Epizootic Hemorrhagic Disease Virus* (EHDV) isolated from Japan and Australia, with nucleotide and amino acid homology of 93.1% to 98.3% and 99.2% to 99.6%, respectively, suggesting that they were Eastern group EHDV. The phylogenetic analysis of outer capsid protein (OC1) and outer capsid protein (OC2) showed that the JC13C644 and JC13C673 viruses were clustered with the EHDV-10 isolated from Japan in 1998, with the nucleotide homology of 98.3% and 98.5%, and the amino acid homology of 99.6% and 99.6–99.8%, respectively, indicating that they belong to the EHDV-10. Seroepidemiological survey results demonstrated that JC13C644 virus-neutralizing antibodies were present in 29.02% (177/610) of locally collected cattle serum and 11.32% (89/786) of goat serum, implying the virus’s presence in Jiangcheng, Yunnan Province. This finding suggests that EHDV-10 circulates not only among blood-sucking insects in nature but also infects local domestic animals in China. Notably, this marks the first-ever isolation of the virus in China and its discovery outside of Japan since its initial isolation from Japanese cattle. In light of these results, it is evident that EHDV Serotype 10 exists beyond Japan, notably in the natural vectors of southern Eurasia, with the capacity to infect local cattle and goats. Therefore, it is imperative to intensify the surveillance of EHDV infection in domestic animals, particularly focusing on the detection and monitoring of new virus serotypes that may emerge in the region and pose risks to animal health.

## 1. Introduction

Epizootic hemorrhagic disease (EHD) in deer is an arboviral illness transmitted by Culicoides. This disease affects a broad spectrum of animals, encompassing domestic cattle, sheep, and various wildlife, including white-tailed deer, elk, and eland. Originally, this condition was identified as “black tongue disease” in the mid-eastern United States during the late 19th century. The first systematic documentation of EHD occurred in 1955 when ailing white-tailed deer were identified in New Jersey, USA, and the EHDV was isolated from these afflicted specimens [1]. EHD manifests in animals through a range of clinical symptoms, including elevated body temperature, respiratory distress, mucosal and serosal bleeding, coma, and in severe cases, fatality. Pregnant cows exposed to EHD may experience miscarriages, fetal malformations, and stillbirths. Notably, the World Organization for Animal Health (WOAH/OIE) designates EHD as a notifiable disease [2,3]. This disease is geographically widespread, occurring between 49° north latitude and 35° south latitude, affecting regions spanning the Americas (particularly the United States), Africa (including Algeria, Tunisia, and Morocco), Europe (in countries like Turkey), and Asia (with instances reported in Japan, Israel, Jordan, South Korea, Taiwan, and China) among others. EHD has recurrently triggered extensive outbreaks within dairy and beef cattle populations, imposing substantial economic losses on the local animal husbandry industry [2,3,4].

EHDV falls within the genus Orbivirus of the family Reoviridae, featuring a viral genome comprised of 10 gene segments in the form of double-stranded RNA (dsRNA). This genome encodes 7 structural proteins (VP1-VP7) and 3 non-structural proteins (NS1-NS3/3a). Notably, two of these 10 gene segments, Seg-2 and Seg-6, are responsible for encoding the virus’s outer membrane proteins, specifically VP2 and VP5. It is worth highlighting that these viral VP2 and VP5 proteins exhibit substantial mutability, consequently allowing for the classification of EHDV viruses into distinct genotypes [5,6,7]. Moreover, Seg-3 serves as the encoding segment for the virus’s inner core protein. Through the molecular genetic evolution analysis of the Seg-3 gene sequence, EHDV viruses can be categorized into various groups, facilitating the elucidation of their geographical distribution characteristics [8].

Presently, researchers have identified seven distinct serotypes of EHDV worldwide, and the geographical distribution of these various serotypes exhibits global variation [9]. Notably, in recent years, a novel serotype of the EHDV virus has emerged in Japan, designated as EHDV-10. Remarkably, this marks the inaugural discovery of a new EHDV serotype beyond the previously known EHDV1-7 serotypes on a global scale [10]. Further investigations have revealed that a diverse range of animals in Japan exhibit positive neutralizing antibodies to this virus. This discovery strongly suggests that the newly identified serotype, EHDV-10, has the capacity to infect a wide spectrum of animals [10].

This study presents the isolation of two viruses closely resembling the recently identified EHDV-10 strain in Japan, obtained from midges collected within Jiangcheng County, Yunnan province. Notably, it underscores the existence of substantial titers of virus-neutralizing antibodies within the local cattle and goat populations. This groundbreaking discovery marks the inaugural instance, outside of Japan, of the EHDV-10 virus being found in its natural habitat and infecting animals. The ensuing findings are detailed below.

## 2. Materials and Methods

### 2.1. Collection of Culicoides Specimens

In August 2013, we conducted the collection of midge specimens within the vicinity of Dazhupeng Village, situated in Menglie Town, Jiangcheng County, Yunnan Province, a region near the borders of China, Laos, and Vietnam (coordinates: East Longitude: 102.0742; North Latitude: 22.4731). This collection process involved nocturnal baiting using light traps (12 V; 300 mA; Wuhan Lucky Star Environmental ProtectionTech Co., Ltd., Wuhan, Hubei, China). Subsequently, the following morning, the insect was promptly placed in a −20 °C refrigerator for 20 min. Insects were taken out from the collection bag, and Culicoides were quickly selected under the stereomicroscope. The species of Culicoides were preliminarily identified according to the size and color of body and, the size and position of pale spots and dark spots on the wings [11]. Every 100 Culicoides with similar morphology collected at the same time in the same place were grouped into one pool and placed in the same cryopreservation tube. These pools were rapidly frozen within liquid nitrogen tanks for preservation before being transported back to the laboratory for further analysis.

### 2.2. Virus Isolation

Culicoides specimens were carefully retrieved from the liquid nitrogen and introduced into a pre-chilled, sterile glass grinder. Each grouping of 100 Culicoides underwent grinding and subsequent centrifugation in accordance with established protocols detailed in the literature [12,13]. Following centrifugation, the supernatant was inoculated into the baby hamster kidney (BHK-21) and *Aedes albopictus* (C6/36) cell. These cultures were meticulously maintained in incubators set at temperatures of 37 °C and 28 °C, respectively. The cell cultures were diligently monitored on a daily basis for the appearance of any cytopathic effects (CPE). Any instances of CPE observed over three consecutive generations were duly categorized as positive outcomes.

### 2.3. Initial Identification of the Virus

To isolate RNA from a 200 μL aliquot of the CPE-positive cell, RNAiso Plus (TaKaRa Company, Dalian Bao Bioengineering Co., Ltd., Dalian, Liaoning, China.) was employed, following the kit instructions, and the extracted RNA was then meticulously preserved at −80 °C. The synthesis of the initial cDNA strand was executed employing the Reverse Transcriptase M-MLV (RNase H-) kit (TaKaRa Company, Dalian Bao Bioengineering Co., Ltd.) in strict accordance with the provided instructions. For the multiplex Polymerase Chain Reaction (PCR) amplification, primers, designed for the conserved regions of prevalent midge-borne arboviruses such as Bluetongue virus, Deer epidemic hemorrhagic fever virus, Palyam serogroup virus, and Simbu serogroup viruses, were employed, drawing reference from pertinent literature [14]. The PCR amplification system employed consisted of 25 μL, comprising 2 μL of cDNA, 12.5 μL of 2× T5 Super PCR Mix, 0.5 μL of upstream and downstream primers, and 9.5 μL of RNase-free ddH_2_O. The amplification process followed specific conditions: initial pre-denaturation at 94 °C for 5 min, followed by 35 cycles of denaturation at 94 °C for 30 s, annealing at 54 °C for 30 s, extension at 72 °C for 60 s, and a final extension at 72 °C for 10 min. Subsequently, the PCR products were subjected to 1% agarose gel electrophoresis in TAE buffer (40 mM Tris-acetate, 1 mM EDTA, pH 8.0) with nucleic acid dye (GoldView) and sequenced by the Shuoqing Genome Biotechnology Co., Ltd., Kunming, Yunnan, China.

### 2.4. Full Genome Sequencing of JC13C644 and JC13C673 Viruses

Following the Full-Length Amplification of cDNAs (FLAC) method outlined as described previously [13,15], we embarked on amplifying the genomes of the JC13C673 and JC13C644 strain viruses. Here is a concise overview of the procedure: 1. Inoculated two positive isolates into BHK21 cells that had just reached a growth area of 75 cm^2^. When approximately 90% of the cells displayed cytopathic effects, the cells were harvested, centrifuged, and total RNA was extracted from the cell pellet using RNAiso Plus (TaKaRa Company, Dalian, China) as per the kit instructions. 2. Single-stranded RNA was eliminated by overnight centrifugation with 2 M LiCl (Sigma, New York, NY, USA) at 4 °C, followed by a 15-min centrifugation at 12,000 rpm. The precipitated RNA was then subjected to further processing. It was mixed with 2.5 volumes of isopropyl alcohol and 1 volume of 7.5 M ammonium acetate, washed twice with 70% ethanol, and eventually dissolved in RNase-free ddH_2_O [16]. 3. Subsequently, 1% agarose gel electrophoresis was employed to separate RNA (run at 7 V/cm for 1 h). Twenty segment bands were excised, comprising S1–S10 for the JC13C673 virus and the JC13C644 virus. These bands were further purified using MiniBEST Agarose Gel DNA Extraction Kit Ver.4.0 (TaKaRa Company, Dalian, China) in accordance with the provided instructions. 4. In accordance with the method outlined by Maan et al., (2007) [17], anchor primers and PC3-T7 loops were attached to the end of each segment (primers synthesized by Shanghai Bioengineering Co., Ltd., Shaihai, China). The ligation reaction was conducted with a 30 μL system, comprising 10 μL of dsRNA (500–1000 ng), 1 μL of RNA ligase (10U) (TaKaRa, Dalian, China), 1.8 μL of BSA, 2.5 μL of PC3-T7 loop (20 μM), 3 μL of DMSO, and 11.7 μL of ddH_2_O. The mixture was thoroughly mixed and subjected to ligation at 15 °C for 16 h. The resulting ligation product was subsequently purified utilizing the MicroElute RNA Cleanup kit (Omega, Guangzhou, China) following the provided procedures. 5. To denature the viral dsRNA ligation product, 15% volume of DMSO was added, and the mixture was briefly heated in boiling water for 2 min followed by a 2-min ice bath. The PrimeScript II high-fidelity RT-PCR kit (TaKaRa) was then employed for the synthesis of the full-length cDNA of the virus’s 10 segments. PC2 primers were utilized to amplify the 10 segmented genes of the virus. 6. The PCR amplification system consisted of 50 μL, including 5 μL of 10× Ex Taq buffer, 5 μL of 2.5 mM dNTPs (TaKaRa), 1.5 μL of primer PC2, 3 μL of cDNA, 0.5 μL of TaKaRa Ex Taq Enzyme (2.5 U), and 35 μL of ddH2O. Prior to the PCR amplification cycles, there was an initial step at 72 °C for 1 min, followed by 95 °C for 5 min. Then, 35 cycles of amplification were performed, involving 94 °C for 20 s, 63 °C for 30 s, and 72 °C for 4 min, concluding with an extension step at 72 °C for 15 min. 7. The resulting amplified product underwent electrophoresis on a 1% agarose gel, and the desired target band was excised and further purified using a gel recovery kit (TaKaRa, Dalian, China). 8. The purified products were ligated into the pMD19-T vector (TakaRa) and subsequently transformed into competent DHα5 cells (TakaRa). Clones were selectively chosen and sent to the Kunming Sequencing Department of Qingke Biotech for sequencing.

### 2.5. Molecular Identification of JC13C644 and JC13C673 Pools Culicoides

The genomic DNA was extracted from JC13C644 and JC13C673 pools Culicoides following the procedure outlined in the literature [18]. Here is a summary of the subsequent steps: 1. We utilized 2 µL of DNA as the template and supplemented it with 0.75 µL of rTaq enzyme, 1.25 µL of upstream and downstream primers for the COI gene [19], 5 µL of each 10× PCR buffer, 5 µL of 2.5 mmol/L dNTPs, and 31.75 µL of DEPC-treated water to compose the PCR reaction mixture. 2. The amplified products underwent thorough scrutiny through 1% agarose gel electrophoresis. Subsequently, the positively detected products were forwarded to the Kunming Sequencing Department of Qingke Biotechnology Co., Ltd. for sequencing analysis.

### 2.6. Sequence Analysis

The sequences of 20 strains of distinct serotypes EHDV downloaded from GenBank were aligned with the corresponding sequences of JC13C644 and JC13C673 viruses using Align in MEGA 7.0 (http://www.megasoftware.net/, accessed on 3 June 2017) [20]. The detailed information for the analysis of virus strain is shown in Appendix A. Homology analysis was performed using MegAlign (DNASTAR Inc.). Sequence assembly and homology analysis of nucleotide and amino acid sequences were performed using SeqMan and MegAlign in DNAStar software (version 4.0; DNASTAR Inc., Madison, WI, USA). The construction of a phylogenetic tree was carried out using MEGA 7.0, employing the maximum likelihood (ML) method [20] with a bootstrap value of 1000 replications.

### 2.7. Detection of Neutralizing Antibodies

#### 2.7.1. Sample Collection

Over the course of five consecutive years, spanning from autumn 2013 to spring 2017, we diligently gathered sera of cattle and goats, within Jiangcheng. This county served as the location where the virus was initially isolated. The collected sera were subsequently transported to the Academy of Yunnan Academy of Animal Husbandry and Veterinary Medicine and safely preserved at −85 °C for further analysis and storage.

#### 2.7.2. TCID50 Assays and Microneutralization Experiments

JC13C644 strain virus was inoculated into BHK21 cells. When 75% of the cells showed obvious CPE, the cell supernatant was harvested, and the TCID50 was determined according to the previously described method [21,22]. Then, the antibody of cattle and goat serum samples collected from Jiangcheng County was detected by microneutralization test. The serum was serially diluted on a 96-well plate, 50 µL per well, and the dilution titer was from 1:4 to 1:1024. Each serum dilution was mixed with an equal volume of virus supernatant (100TCID50/50 µL). Each serum dilution was repeated four wells, and viral supernatants of 100TCID50, 10TCID50, 1TCID50, and blank control were set up as the controls, respectively. The plates were incubated in a 5% CO2 incubator at 37 °C for neutralization, and then 100 µL BHK-21 cell suspension was added to each well. The plates were cultivated in a 5% CO_2_ incubator at 37 °C for 7 days. The serum was determined as virus-positive when the neutralizing antibody titer was >1:16.

## 3. Results

### 3.1. Specimen Collection and Virus Isolation

In 2013, within the vicinity of Dazhupeng Village, situated in Jiangcheng County, Yunnan Province, China, near the border shared by China, Laos, and Vietnam, our research team diligently collected *Culicoides* specimens. This comprehensive effort yielded a total of 15,000 Culicoides, encompassing nine distinct species, including *Culicoides (Trithecoides) humeralis, Culicoides (Trithecoides) matsuza, Culicoides (Avaritia) tainanus, Culicoides (Avaritia) sumatrae, Culicoides (Avaritia) actoni, Culicoides (Avaritia) liui, Culicoides (Avaritia) jacobsoni, Culicoides (Avaritia) insignipennis, Culicoides (Oecacta) oxystoma, and Culicides (Beltranmyia) arakawae*. Notably, *Culicoides humeralis* (comprising 28%) and *Culicoides tainanus* (constituting 25.5%) emerged as the predominant species among the collected Culicoides. To facilitate our research, the 15,000 Culicoides specimens were meticulously divided into 150 pools for processing. These pools were then subjected to grinding and subsequent inoculation with BHK-21 cells and C6/36 cells for the purpose of virus isolation. Two strains of virus (with the same strain number and batch number) were isolated from two batches of midge specimens, JC13C673 and JC13C644 (JC refers to the location of the midge collection site, Jiangcheng; 13 refers to the time of collection of the specimens from which the virus was isolated; 673 and 644 are the midge pool numbers). Therefore, we named the isolated viruses JC13C673 and JC13C644. Remarkably, both the JC13C673 and JC13C644 viruses exhibited unmistakable CPE within a mere 36 h post-inoculation into BHK21 cells. These effects were characterized by pronounced cellular rounding, detachment, and fragmentation.

### 3.2. Biological Characteristics of Both the JC13C644 and JC13C673 Strains

The two virus strains, JC13C644 and JC13C673, exhibited similar CPE upon inoculation with BHK21 cells. These effects were characterized by pronounced cellular rounding, shedding, and fragmentation. Notably, the onset of CPE occurred at 40 h for JC13C644 and 36 h for JC13C673, respectively. Furthermore, these two virus strains displayed comparable plaque morphologies. Specifically, the plaques formed by the JC13C644 strain exhibited a diameter of 4.5 ± 0.67 mm (*n* = 10), while those formed by the JC13C673 strain had a diameter of 4.2 ± 0.72 mm. Subsequently, the virus titers for both strains after inoculation into BHK21 cells were determined to be 5.2 × 10^6^ and 6.5 × 10^6^ PFU/mL, respectively. Interestingly, when C6/36 cells were inoculated with these virus strains, no observable CPE emerged even after 7 days. However, evident CPE appeared within 3 days of inoculation with Vero cells, and after 7 days of inoculation with MDBK cells. Furthermore, upon intracerebral inoculation into suckling mice, both virus strains elicited symptoms such as milk rejection, nest abandonment, tremors, and convulsions within 2 days, ultimately leading to the demise of the suckling mice. However, inoculation into the embryonic veins of 10-day-old chickens did not result in the mortality of chicken embryos, as indicated in Table 1.

### 3.3. Molecular Biological Identification of Two Pools of Culicoides Samples JC13C673 and JC13C644

Two pools of *Culicoides* JC13C673 and JC13C644 were subjected to PCR amplification using the COI gene-specific primers. The 500 nt COI gene sequence was obtained by sequencing. Phylogenetic analysis showed that two pools of Culicoides JC13C673 and JC13C644 were located in the same evolutionary branch with *C. tainanus* or *C. maculatus* (*C. tainanus* and *C. maculatus* are the same species of *Culicoides*) (Figure 1), and the nucleotide homology was 95%, indicating that the species of JC13C673 and JC13C644 pools were identified as *C. tainanus*.

### 3.4. Virus Identification

We initiated multiplex RT-PCR amplification by employing specific primers tailored for the bluetongue virus, EHDV, Paya serogroup virus, and Akabane virus. Our results unequivocally demonstrated positive amplification for EHDV in the case of the two strains, JC13C644 and JC13C673. Subsequently, the nucleotide sequence of approximately 600 nt was successfully determined. Through Blastn, these two virus strains exhibited the highest nucleotide homology (98%) in seg-3 with the EHDV strain ON-4/B/98, which was previously isolated in Japan.

### 3.5. Full Genome Sequence Determination and Analysis

The FLAC method was meticulously employed to sequence the complete genomes of the recently isolated viruses, JC13C644 and JC13C673, yielding the gene sequences of segments 1–10, denoted as Seg, for both viruses. These sequences exhibited a notable range in fragment length, spanning from 3942 base pairs (Seg-1) to 810 base pairs (Seg-10). Detailed information regarding the specific length of each gene fragment, the corresponding encoded protein length, and their respective gene numbers can be found in Table 2. Furthermore, a comprehensive sequence analysis of the 5′ and 3′ non-coding regions (NCRs) of each gene fragment unveiled a striking pattern of conservation. Specifically, it was observed that the 6 nucleotides located at the 5′ end and the 3 nucleotides positioned at the 3′ end of all 10 virus fragments exhibited a high degree of conservation, manifesting as “5′GUUAAA- --UAC3′”. Additionally, the first two nucleic acids from the 5′ end and the last two nucleic acids from the 3′ end were found to be reverse complementary, further highlighting the consistency in this crucial region across the viral genome. Compared with the EHDV-10 ON-4/B/98 strain, the JC13C644 and JC13C673 virus has 6 nt insertions in the Seg-8 open reading frame (777-782). This similar insertion was also observed in EHDV5 _ CSIRO _ 157 strain _ (AM745034.1) and EHDV8 _ CPR _ 3961A strain (AM745064.1).

### 3.6. Nucleotide and Amino Acid Identity Analysis of the Two EHDV Virus Strains

The two virus strains, JC13C644 and JC13C673, displayed the highest degree of homology with the Japanese EHDV-10 isolate ON-4/B/98, specifically in segments Seg-1, Seg-2, Seg-3, Seg-4, Seg-5, Seg-6, Seg-7, Seg-9 and Seg-10, with 97.5–99% at nucleotide and 96.7–100 at amino acid, while Seg-8 had low homology between them, with 70–70.1% at nucleotide and 75.7% and 75.9% at amino acid. In contrast, when compared to other serotypes in Seg-2, nucleotide, and amino acid homology exhibited significant variations, ranging from 44.5% (EHDV-6) to 69.4% (EHDV-2) at the nucleotide level, and from 32.6% (EHDV-6) to 68% (EHDV-2) at the amino acid level. For Seg-3, nucleotide homology spanned from 79.4% (EHDV-1) to 97% (EHDV-8), while amino acid homology ranged from 95.3% (EHDV-6) to 99.4% (EHDV-8). For Seg-6, nucleotide homology ranged from 60.7% (EHDV-5) to 80.7% (EHDV-7), while amino acid homology varied from 62% (EHDV-5) to 95.8% (EHDV-2). In the case of Seg-1, Seg-4, Seg-5, Seg-7, Seg-8, Seg-9, and Seg-10, the nucleotide homology ranged from 77.2% (EHDV-1) to 95.7% (EHDV-8), and amino acid homology within these segments spanned from 89.5% (EHDV-1) to 99% (EHDV-8). For further detailed comparisons, please refer to Table 3.

### 3.7. Analysis of Genetic Evolution of Seg-2, Seg-3 and Seg-6 of the Two EHDV Strains

#### 3.7.1. The Phylogenetic Analysis of Segments Seg-1, Seg-3, Seg-4, Seg-5, Seg-7, Seg-8, Seg-9, and Seg-10 from JC13C673 and JC13C644 Viruses Was Conducted in Comparison with EHDV-10 and EHDV-1 to EHDV-8

The sub-core-shell protein (T2) encoded by segment 3 (Seg-3) of JC13C673 and JC13C644 is VP3. The phylogenetic tree based on Seg-3 of JC13C673 and JC13C644 exhibits a close genetic relationship to EHDV strains prevalent in Eastern countries like Japan and Australia (Figure 2A). This relationship is supported by a high nucleotide homology of 93.1% to 98.3% and amino acid homology of 99.2% to 99.6%. However, the genetic link with EHDV strains common in Western countries such as Israel, Kenya, and the United States is more distant, with nucleotide and amino acid homologies below 90% and 97%, respectively (Table 3). These findings suggest that both viruses isolated from Culicoides in the China-Laos-Vietnam border area belong to the Eastern group EHDV group. Similar results were observed in the phylogenetic tree analysis of six additional gene segments: S1, S4, S5, S8, S9, and S10 (Appendix A–G). These segments also place the two EHDV strains from Culicoides in the China-Laos-Vietnam border area within the same evolutionary lineage as Eastern group EHDV viruses prevalent in Eastern countries such as Japan and Australia.

#### 3.7.2. Phylogenetic Analysis of S2 and S6 of JC13C673 and JC13C644 Virus Compared with EHDV10 and EHDV1-8

Segments S2 and S6 encode the outer capsid proteins VP2 and VP5, respectively. In the phylogenetic tree constructed for the Seg-2, all EHDV types 1-10 can be grouped into four major evolutionary branches: Group A to D. Notably, the newly isolated JC13C673 and JC13C644 viruses fall within a significant evolutionary branch, specifically Group A, alongside EHDV-10, EHDV-2, and EHDV-7 (Figure 2B). Upon closer examination, it is evident that these newly isolated viruses, JC13C673 and JC13C644, share a striking similarity with the EHDV-10 strain isolated in Japan in 1998. They are placed in the same evolutionary cluster, displaying a nucleotide homology of 97.6% to 97.7% and an amino acid homology of 98.1% to 98.2%. In contrast, they are situated in a distinct evolutionary cluster from other serotype viruses, with nucleotide and amino acid homologies consistently below 70% (Table 3). The results of the phylogenetic tree constructed for the S6 gene closely mirror those of the Seg-2. The two newly isolated viruses, JC13C673 and JC13C644, are positioned within the Group A evolutionary branch (Appendix A). Specifically, these viruses share an evolutionary lineage with the EHDV-10 virus isolated in Japan in 1998 but maintain a distinct separation from other serotype viruses.

### 3.8. Survey on Serum Antibodies of Domestic Animals

EHDV-10 virus was initially isolated from animals in Japan back in 1998. However, there remains a lack of clarity regarding the pathogenicity and epidemiological status of this virus. In 2013, our study successfully isolated two strains of EHDV-10 from Culicoides in Yunnan, China. To gain insight into the prevalence and infection dynamics of this virus in local animal populations, we conducted a comprehensive survey in seven towns within Jiangcheng over a span of five years (2013–2017). This survey encompassed a total of 1,396 serum samples from both goats and cattle, including 610 cattle and 786 goats. We employed the microneutralization test method to detect antibodies against the EHDV-10 virus, considering a titer equal to or greater than 1:16 as indicative of a positive result. The overall positive rate for EHDV-10 viral antibodies was calculated at 19.05% (266/1396). When stratified by species, the positive rate for viral antibodies in cattle serum stood at 28.52% (174/610), while the positive rate in goat serum was 11.70% (92/786). The highest neutralizing antibody titers recorded for cattle and goats exposed to EHDV-10 were 1:1024 and 1:256, respectively. Furthermore, the positive rates for neutralizing antibodies with titers exceeding 1:64 were found to be 15.25% (93/610) for cattle and 0.89% (7/786) for goats. From a geographical perspective, it is noteworthy that Qushui Town exhibited the highest positive rates for EHDV-10 virus-neutralizing antibodies in cattle and goats at 41.18% and 16.92%, respectively. Conversely, Zhengdong Town had the lowest positive rate for the virus in cattle serum at 20%, and Menglie Town recorded the lowest viral antibody positivity in goat serum at 6.88% (Figure 3A,B). Examining the temporal dimension, the positive rates for virus-neutralizing antibodies in cattle serum were 10% (2013), 26.79% (2014), 40.91% (2015), and 33.33% (2017). For goat serum, the corresponding antibody positivity rates were 1.79% (2013), 11.45% (2014), 10.86% (2015), 13.39% (2016), and 31.25% (2017).

## 4. Discussion

The distribution of EHDV serotypes globally exhibits distinct regional patterns. In North America, serotypes 1, 2, and 6 are prevalent [23,24,25], whereas in Africa, serotypes 1, 2, 3, 4, and 6 [26,27,28] are found. In Australia, serotypes 1, 2, and 5 [29,30] are documented. In Asia, specifically Japan, serotypes 6, 7, and 8 are prominent, alongside serotypes 1, 2, 4, 6, and 7 [10,31,32]. Meanwhile, in China, serotypes 1, 5, 6, and 7 [33,34] are observed. A significant development in EHDV research occurred in 2017 when a novel serotype, EHDV-10, was isolated from cattle blood in Japan, denoted as ON-4/B/98 [10]. This strain exhibited distinct genetic evolution characteristics compared to all previously known serotype 1–7 viruses. Serological tests revealed that the ON-4/B/98 virus strain resisted neutralization by antibodies from other EHDV serotypes. Consequently, it was designated as the EHDV-10 virus. This marked not only a pioneering discovery in Japan but also globally, as it represented the first report of a new EHDV serotype beyond the established EHDV1-7 serotypes. In recent years, the EHDV-10 virus has been recurrently isolated from sentinel surveillance animals in the Yunnan border area [35]. In the present study, we isolated two virus strains in Jiangcheng County in 2013. The nine segments of the two newly isolated viruses except Seg-8 had the same length, open reading frame size, conserved terminal nucleotide sequences of the 5′ and 3′ UTRs, and comparable G+C% content with the EHDV-10 ON-4/B/98 strain isolated in Japan. Additionally, phylogenetic analysis of the nine segments indicated that the two newly isolated viruses share the closest genetic relationship with EHDV-10 ON-4/B/98, demonstrating a nucleotide homology of 97.5–99% and an amino acid homology of 96.7–100%. This strongly suggests that the two newly isolated viruses in the China-Laos-Vietnam border area are closely genetically related to EHDV-10 ON-4/B/98. Further genetic evolution assessments of Seg-2 and Seg-6 reinforced this conclusion, revealing a close genetic association with the EHDV-10 ON-4/B/98 strain. The amino acid homology for both segments exceeded 98% (VP2) and 99% (VP5). Particularly noteworthy is the substantially higher amino acid homology in the VP2 protein between these viruses compared to most EHDV serotypes (76.7% being the highest recorded). Consequently, this strongly suggests that both newly isolated viruses from the China-Laos-Vietnam border area are likely EHDV-10 strains.

The outer capsid layer, comprised of VP2 and VP5 proteins, plays a crucial role in initiating cell infection and facilitating the delivery of the viral core into the cytoplasm, thereby triggering the essential biochemical processes of replication [7]. Among the genome segments of EHDV, Seg-2, which encodes VP2, is the most variable, while Seg-6, encoding VP5, ranks as the second most variable segment. Based on the genetic evolution analysis of amino acid sequences within Seg-2 and Seg-6, all known EHDV strains are categorized into four distinct groups, labeled as A, B, C, and D. Group A encompasses EHDV2, EHDV7, and EHDV10; group B includes EHDV6 and EHDV8; group C comprises EHDV1, and group D consists of EHDV4 and EHDV5. The genetic evolution analysis of Seg-2 and Seg-6 conducted in this study establishes that the two newly isolated viruses belong to group A, sharing this group with viral strains EHDV2, EHDV7, and EHDV10. Notably, the nucleotide homology between the two newly isolated viruses and EHDV2, EHDV7, and EHDV10 exceeds 67.5% (S2) and 78.7% (S6), while the amino acid homology surpasses 65.2% (S2) and 91.9% (S6). In contrast, their nucleotide homology with EHDV strains from groups B, C, and D is less than 48.8% (S2) and 69.1% (S6), with amino acid homology below 35.4% (S2) and 78.4% (S6). These findings affirm the close genetic relationship between the two newly isolated viruses EHDV2 and EHDV7.

The T2 protein, a component of the Orbivirus genus, stands out as the most conserved among its genes and is closely associated with viral speciation [36]. This T2 protein, encoded by Seg-3, has been widely employed to investigate the geographical genetic variations (topotypes) within related EHDVs and BTVs. In the context of the genetic evolution analysis of Seg-2 nucleotide sequences conducted in this study, it is evident that the two newly isolated viruses share a remarkably close genetic evolutionary relationship with Eastern group EHDV strains sourced from Japan, China, Australia, and various other countries. This relationship is further underscored by their notably high nucleotide and amino acid homology, reaching 93.1% and 99.2%, respectively. These findings strongly suggest that the two viruses isolated within the China-Laos-Vietnam border region are indeed Eastern group EHDV viruses. Furthermore, the genetic evolution analysis results obtained for six gene segments, encompassing S1, S4, S5, S8, S9, and S10, consistently indicate that these two newly isolated strains also belong to the Eastern group EHDV category. This concurrence underscores the alignment between the geographical distribution of these newly isolated virus genotypes and the broader distribution pattern of EHDV within the Asia-Pacific region.

Some studies have posited the existence of subtle distinctions in the characteristics of different serological strains, particularly with regard to Seg-2/VP2, Seg-6/VP5, and Seg-3/VP3 in circulation within Japan [10]. In this study, two freshly isolated virus strains were examined, specifically within the SEG-2/VP2 category. While the fragment length and open reading frame size of EHDV-2 and EHDV-7 in Group A align, noteworthy distinctions manifest in protein molecular weight, G+C content, and the sequences at the 5′ and 3′ ends. Concerning Seg-6/VP5, the length of the fragments from these two newly isolated viruses matches that of EHDV-8 but diverges from other serotype viruses. However, the reading frame length coincides with those of other serotype viruses, except for EHDV4 and EHDV5. Variations in G+C content and 5′ and 3′ end sequences are observed. In Seg-5/NS1, the fragment length of the two newly isolated viruses mirrors that of EHDV2, EHDV5, and EHDV7, standing apart from other serotype viruses. Nonetheless, the reading frame lengths remain consistent across serotypes. Moving on to Seg-8/NS2, the fragment length and reading frame length for the two newly isolated viruses correspond to those of EHDV5, EHDV7, and EHDV-8, in contrast to other serotype viruses. Differences are detected in G+C content and 5′ and 3′ end sequences. For Seg-9/VP6, the fragment length of the two newly isolated viruses matches that of EHDV2, prevalent in Japan, while differing from other viruses. Correspondingly, the reading frame length is shared with EHDV2 and EHDV5, along with EHDV7 (YN09), distinguishing them from other viruses. G+C content and 5′ and 3′ end sequences also exhibit variations. Among the remaining five fragments, the fragment length and reading frame align with EHDV strains prevalent in the Asia-Pacific region. Nevertheless, there are discernible differences in G+C content and 5′ and 3′ end sequences. Notably, the 10 segments of these two newly isolated viruses exhibit six conserved nucleotides at the 5′ end (5′GUUAAA) and three conserved nucleotides at the 3′ end (-GAC 3′). Additionally, the last two nucleotides at the 3′ end complement the first two nucleotides at the 5′ end. This pattern mirrors that found in other EHDV viruses, establishing a shared motif structure at the 5′ and 3′ ends of the virus.

The primary susceptible animals to EHDV are deer and cattle. Interestingly, in 2015, the presence of goat EHDV antibodies (9.2%) was detected in goats, which are under surveillance in Guangxi [37]. This discovery suggests that goats may also be susceptible to EHDV infection. In this study, extensive screening for neutralizing antibodies specific to the newly isolated EHDV-10 was conducted in cattle and goats across seven towns in Jiangcheng, the very area where the virus was originally isolated. The results revealed that the positive rate of neutralizing antibodies to the newly isolated EHDV-10 virus, at a titer of ≥1:16, was 28.52% in cattle. Conversely, in goats, the positive rate was 11.70%. Particularly noteworthy was the observation that among cattle with a neutralizing antibody titer of ≥1:64 to the newly isolated virus, the positive rate remained remarkably high at 15.25%, with the highest neutralizing antibody titer recorded at 1:1024. In contrast, the positive rate in goats was only 0.89%, and the highest neutralizing antibody titer was 1:1024, while the overall antibody titer in goats reached only 1:256. This evidence strongly suggests that both cattle and goats in Jiangcheng have been exposed to the newly isolated EHDV-10. However, cattle appear to be more susceptible to the newly isolated virus, with a more significant prevalence of EHDV-10 infection observed among local cattle. 

Upon conducting further analysis, it was revealed that the positive rates for cattle in the seven townships within Jiangcheng ranged from 20% to 41.18%, while for goats, these rates varied from 6.88% to 16.92%. This compelling data strongly suggests that the newly isolated EHDV-10 virus has established a pervasive presence among both cattle and goat populations in these seven towns within Jiangcheng, confirming the extent of infection. When considering the time frame of sample collection (spanning 2013–2017), the positive rate of virus-neutralizing antibodies in bovine serum was observed to range from 10% (in 2013) to 40.91% (in 2015). Although data for bovine serum samples from 2016 is regrettably unavailable, the positive rate of virus-neutralizing antibodies in goat serum over the past five years ranged from 1.79% (in 2013) to 31.25% (in 2017). Notably, the positive rates in both cattle and goat serum samples collected during spring and autumn were higher, at 20.59% and 13.83%, respectively. This indicates that EHDV-10 infection remains prevalent among cattle and goats in Jiangcheng throughout the year, with no distinct seasonal patterns. This suggests that EHDV infection is persistent and actively circulating among cattle and goats in Jiangcheng.

Considering the potential cross-reactions between EHDV, BTV, and other orbiviruses, it is crucial to note that the neutralizing antibody titers for the EHDV-10 virus in local cattle exhibited a range from ≥1:16 to <1:64. This resulted in a positive rate of 13.27% among cattle and 10.81% among goat. Furthermore, it is important to mention that our laboratory has also successfully isolated multiple strains of the EHDV-7 virus from cattle in Jiangcheng. These compelling findings indicate that cattle and goats in Jiangcheng, apart from being exposed to EHDV-10 isolated from Culicoides, might potentially face infections from other related serotypes of EHDV.

EHD, an animal viral disease, is primarily transmitted by blood-sucking midges. The occurrence, prevalence, and geographical distribution of this disease are intricately linked to the species and geographical spread of its primary transmission vector, Culicoides [3]. Current research highlights *C. obsoletus*, *C. actoni*, *C. brevitarsis*, *C. imicola*, and *C. oxystoma* as the primary vectors or potential carriers of EHDV [38,39]. In this study, two strains of EHDV-10 were successfully isolated from *C. tainanus* collected from cattle pens in Jiangcheng. This discovery strongly suggests that *C. tainanus* could serve as a potential transmission vector for local EHDV-10. Among the nine Culicoides species collected during this study, *C. tainanus* (comprising 25.5% of the specimens) emerged as the dominant species at the collection site. This finding underscores the significant role that *C. tainanus* might play in the natural cycle of local EHDV-10.

EHD is a global concern, with notable regions of prevalence spanning North America, Africa, Australia, and Asia [29]. In recent times, this disease has extended its reach into the Mediterranean Basin, affecting countries like Morocco, Algeria, Tunisia, Israel, Jordan, and Turkey [38]. This expansion demonstrates that EHDV has not only established a wide geographical presence but continues to broaden its scope. This study focused on the isolation site of Jiangcheng, Yunnan Province. Remarkably, this location is merely about 5 km away from the border in a direct line. The area is characterized by low altitude, abundant rainfall, high humidity, and extensive forest cover. In-depth vector insect surveys conducted in this region revealed a high Culicoides density, including several species like *C.tainanus, C.actoni, and C.oxystoma*, which act as EHDV transmission vectors or potential reservoirs. Importantly, these Culicoides vectors are present in the region every December (based on unpublished data). The serological survey of local animals in this study further corroborated these findings, as it detected EHDV10-neutralizing antibodies in cattle and goat serum collected during both spring and autumn. This suggests that cattle and goats in this area may face year-round EHDV10 infections due to Culicoides bites, highlighting the persistent nature of this issue.

This study marks a pivotal moment, as it signifies the inaugural isolation of the EHDV10 from midges in this region. This achievement has shed light on the genetic connections between the EHDV-10 strains found in Chinese midges and those prevalent in Japan, Australia, and other locales. Additionally, it has substantiated the widespread prevalence of EHDV10 infections in local cattle and goat populations, affirming the virus’s natural circulation. However, the question of whether this virus poses a threat to domestic animals such as cattle and goats necessitates further intensive monitoring and research efforts. These efforts should focus on diseases induced by the EHDV10 in local cattle and goat populations. The data gleaned from this study will serve as a valuable resource in enhancing the diagnosis and prevention of EHD within China and neighboring regions.

## 5. Conclusions

This study successfully isolated two strains of the EHDV10 from *C. tainanus* collected in Jiangcheng County, Yunnan province. Both virus strains exhibited the capacity to induce CPE in various mammalian cell lines (BHK, Vero, MDBK), resulting in morbidity and mortality in suckling mice. Furthermore, findings from the seroepidemiological investigation suggest that the EHDV10 is not only present in natural blood-sucking insects within the China-Laos-Vietnam border areas but also infects local domestic animals. Remarkably, this marks the frst time that a virus of this serotype has been isolated from the vector Culicoides, subsequent to its discovery in animals in Japan. These outcomes signify the existence of the EHDV10 beyond Japan, demonstrating that the virus naturally occurs in southern Eurasia and infects domestic animals like cattle and goats in the region. Given these findings, it becomes increasingly imperative to intensify the detection and surveillance of animal diseases caused by EHDV.

## Figures and Tables

**Figure 1 viruses-16-00175-f001:**
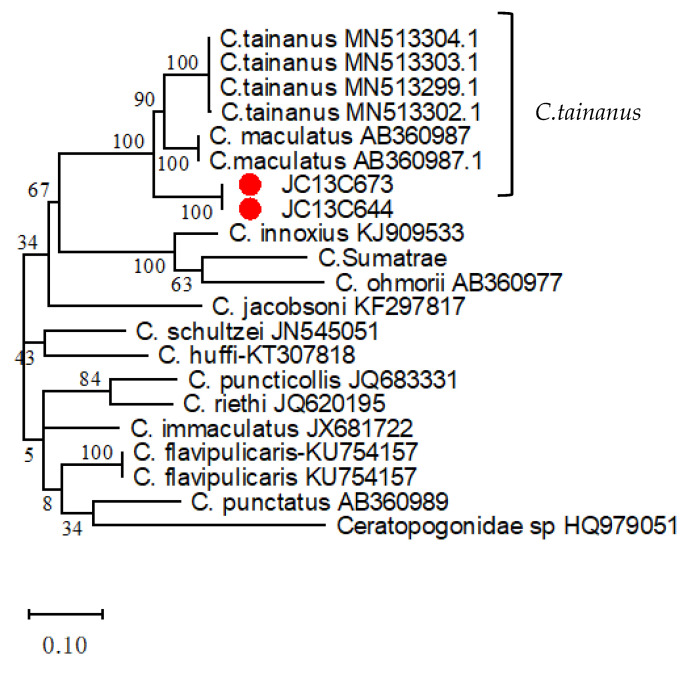
ML phylogenetic trees of COI gene of *C. tainanus* JC13C673 and JC13C644 pools. The best substitution model is GTR+G; The bootstrap value is 1000.

**Figure 2 viruses-16-00175-f002:**
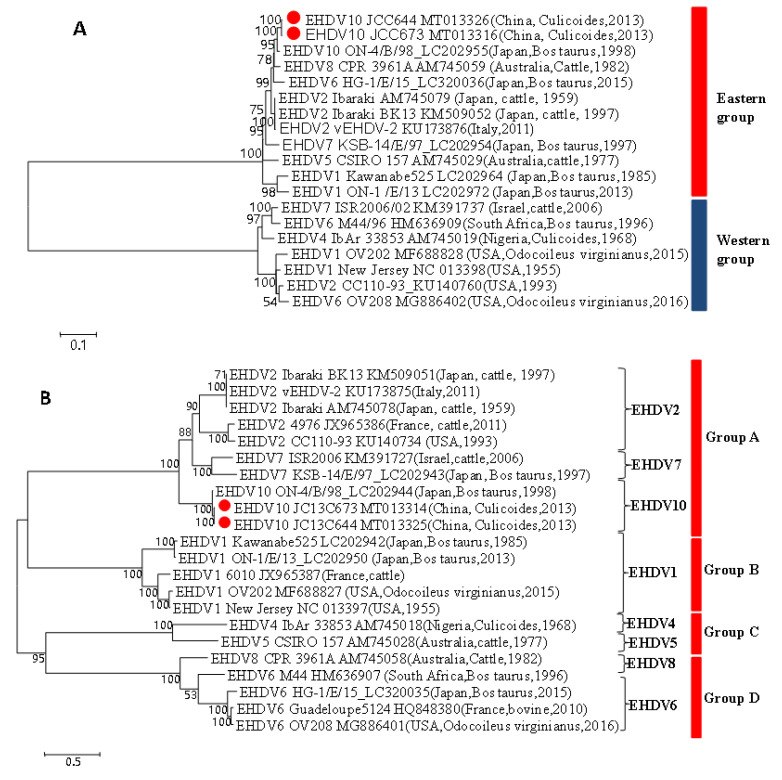
ML phylogenetic trees of Seg-3 (VP3) (**A**) and Seg-2 (VP2) (**B**)of JC13C673 and JC13C644 strains EHDV isolated from Jiangcheng county. The best substitution model is GTR+G+I (Seg-3) and TN93+G+I (Seg-2); The bootstrap value is 1000. ● The red dots indicated the newly isolated virus strain in this study.

**Figure 3 viruses-16-00175-f003:**
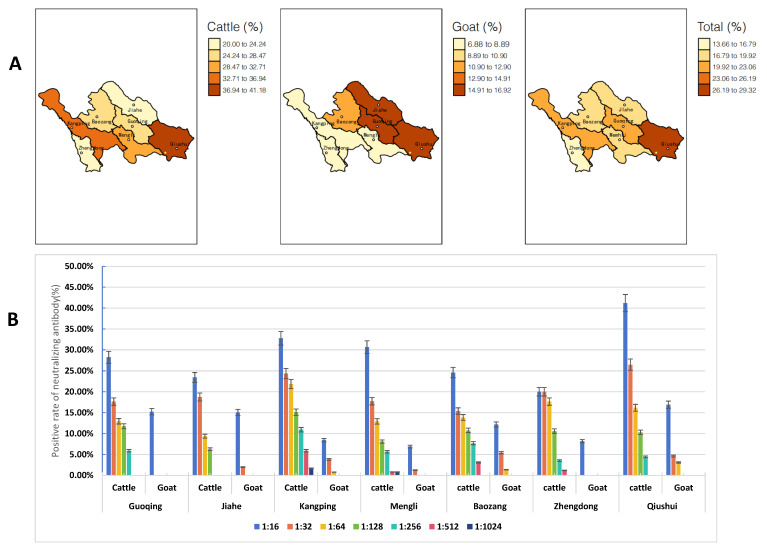
The detection of EHDV-10 neutralizing antibody in cattle and goat in seven townships of Jiangcheng County, Yunnan Province. (**A**) The geographical distribution of neutralizing antibodies; (**B**) The distribution of neutralizing antibody titers in different regions and different animals.

**Table 1 viruses-16-00175-t001:** Biological characteristics of virus JC13C644 and JC13C673 isolated from *C. tainanus* in China-Lao-Vietnam border area.

		JC13C673	JC13C644
CPE time (Hour)	BHK21	32	40
	Vero	66	72
	MDBK	168	168
	C6/36 *	-	-
Animal lethality	chick embryo	No deaths	No deaths
	suckling mice	Death 48 h after inoculation	Death 48 h after inoculation
virus titer		5.2 × 10^−6^	6.5 × 10^−5^
Plaque size (diametermm)		4.2 ± 0.72	2.5 ± 0.97

*: means that no obvious CPE was observed for 7 consecutive days.

**Table 2 viruses-16-00175-t002:** Lengths of dsRNA segments 1–12, encoded putative proteins, 5′ and 3′NCRs of EHDV (JCC13C673 and JC13C644) genome.

Segment	Length (bp)	Protein (aa)	5′NCR (bp)	G+C Content (mol%)	Terminal Sequence (5′–3′)	3′NCR (bp)	GenBank Accession No.	
							JC13C644	JCC13C673
S1	3942	1302	11	39.55	**GUUAAA**AUGC------------CACU**UAC**	22	MT013324	MT013314
S2	3002	982	17	39.71	**GUUAAA**UUGT------------AACU**UAC**	36	MT013325	MT013315
S3	2768	899	17	42.99	**GUUAAA**UUUC------------CACU**UAC**	51	MT013326	MT013316
S4	1983	644	8	41.05	**GUUAAA**ACAU-------------CACC**UAC**	40	MT013327	MT013317
S5	1769	551	31	42.62	**GUUAAA**AAGU------------CACU**UAC**	82	MT013328	MT013318
S6	1642	527	27	43.42	**GUUAAA**AAGG------------CACU**UAC**	31	MT013329	MT013319
S7	1162	349	17	43.55	**GUUAAA**AUUU------------AACU**UAC**	95	MT013330	MT013320
S8	1192	375	19	45.22	**GUUAAA**AAUU-----------ACACU**UAC**	45	MT013331	MT013321
S9	1075	337	15	44.93	**GUUAAA**AAUU------------AACU**UAC**	46	MT013332	MT013322
S10	810	228	20	44.44	**GUUAAA**AAGA------------CACU**UAC**	103	MT013333	MT013323
		Consensus			**GUUAAAWW**--------------**RMMYUAC**			

**Table 3 viruses-16-00175-t003:** Comparison of each segment between virus JC13C673, JC13C644 and other EHDV in nucleotide and amino acid identities.

Segment	EHDV1 *		EHDV2 ^#^		EHDV4 ^$^		EHDV5 ^£^		EHDV6 ^§^		EHDV7 ^¥^		EHDV8 ^@^		EHDV10 ^&^	
	nt(%)	aa(%)	nt(%)	aa(%)	nt(%)	aa(%)	nt(%)	aa(%)	nt(%)	aa(%)	nt(%)	aa(%)	nt(%)	aa(%)	nt(%)	aa(%)
S1(VP1)	77.2–77.9	89.5–90.2	77.7–94.7	89.8–98.9	78.4–78.5	90.3–90.5	93.5–93.6	97.7–97.8	77.4–78.3	89.6–90.9	78.2–93.3	90.9–98.6	95.6–95.7	98.8–99	98–98.2	99.1–99.2
S2(VP2)	47.4–48.8	34.6–35.4	67.6–69.4	67.3–68	46.5–46.6	33.2–33.4	46.4	34.4–34.6	44.5–45.8	32.6–35	67.5–68.9	65.2–67.1	45.8–45.9	33.5–33.7	97.6–97.7	98.1–98.2
S3(VP3)	79.4–91.8	95.4–99.2	80.1–96.5	95.6–99.4	79.5	96.4	93.1	99.3	79.6–95	95.3–99.4	80–96.4	96.2–99.6	97	99.4	98.3	99.6
S4(VP4)	74.9–75.4	84.8–85.2	75.7–93.4	84.8–98.6	76.2	85.6–85.7	92–92.1	97.4–97.5	74.7–76.1	85.4–85.7	75.6–95.3	84.8–98.6	95.6–95.8	98.4–98.6	97.5–97.8	98.9–99.1
S5(NS1)	79.1–79.6	91.8–92.6	79.3–98	92–99.3	79.5–79.7	92.6–92.7	93.7–94	98.5–99.1	78.5–79.4	91.8–92.2	78.7–97.1	91.5–99.3	86.3–86.6	97.5–98	98.1–98.3	99.3–99.8
S6(VP5)	63.4–66.2	67.3–67.5	78.7–79.3	91.9–95.8	61.4	62.6	60.7–60.8	62	67.2–69.4	77.1–78.4	79.7–80.7	92.2–93.8	69–69.1	77.5–77.8	98.5	99.6–99.8
S7(VP7)	77.5–79.9	94.9	83.9–84.4	98–98.9	77.3	92.3	79.8	94.3	81.2–82.7	97.7–98.3	82.1–84.5	96–98.9	79.5	95.5	98.4	100
S8(NS2)	69.1–71.3	73.9–74.4	70.3–71.1	73.3–74.1	70.1–70.3	73.1	79.4–79.6	85.9–86.1	67.7–70.3	72.3–73.9	71.2–96.2	72–96	79.5	86.1	70–71.1	75.7–75.9
S9(VP6)	67.5–70.6	63.7–64.8	71.1–91.1	64.8–92	70.5–70.6	62.4–62.9	94.4–94.7	93.7–94.2	68.8–71.2	63.5–64.3	70.7–94	62.6–93.2	95.9–96.2	95.6–96.4	98.9–99	97.3–96.7
S10(NS3)	74–76.3	86.4–87.3	77.2–92.3	87.3–98.7	80.2	92.5	94.2–94.4	99.6	74.4–76.3	86.8–87.3	75.7–77	84.6–87.3	77.4–77.7	84.6	98.2–98.4	99.6

Note: ***** EHDV1:_OV202_MF688834.1, New_Jersey_NC_013403.1; **^#^** EHDV2:_Ibaraki_BK13_KM509057.1, Ibaraki_AM745084.1, EHD2/USA1993/AL/CC110-93_KU140856.1, vEHDV-2_Ibaraki_KU173881.1; **^$^** EHDV4:_IbAr_33853_AM745024.1; **^£^** EHDV5:_CSIRO_157_AM745034.1; **^§^** EHDV6:_OV208_MG886408.1, M44/96_HM636914.1; **^¥^** EHDV7:_ISR2006/02_KM391745.1; **^@^** EHDV8:_CPR_3961A_AM745064.1; **^&^** EHDV10_ON-4/B/98_LC552752.1.

## Data Availability

The nucleotide sequences of the complete genome sequence of JC13C644 and JC13C673 strains obtained in this study were submitted to the GenBank database under accession numbers (MT013314-MT013333). The background information and GenBank accession numbers of all virus strains used in this study.

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
