# Peer review of "Isolation of Epizootic Hemorrhagic Disease Virus Serotype 10 from Culicoides tainanus and Associated Infections in Livestock in Yunnan, China"

_viruses, 2024, doi:10.3390/v16020175_

Round 1
Reviewer 1 Report
Comments and Suggestions for Authors
General comments
Epizootic hemorrhagic disease (EHD) is a WOAH-listed disease and can cause significant economic losses in ruminant livestock production. This study by He et al. is the first report of isolation of a provisional serotype of EHD virus (EHDV) serotype 10 (EHDV-10) outside Japan. This study provides new insights into the distribution of EHDV, but the manuscript contains many rudimentary errors that may confuse readers and should be corrected.
Specific comments
Abstract
Lines 17-21
The second, third, and fourth sentences are unnecessary; they may be important if the timing of CPE appearance, plaque sizes, and viral titer are clearly different from that of other EHDV strains. Otherwise, they are unlikely to need to be stated.
Line 28
Are there any other examples of the expression “Oriental genotype” used for EHDV? I do not think this expression is appropriate. Please correct it, as well as other parts of the manuscript
Line 34
The first sentence of the Abstract indicates that Jiangcheng borders Laos and Vietnam. It is acceptable to mention once that Jiangcheng borders Laos and Vietnam as information about the location where the research was conducted, but it is very unnatural to mention "the border area shared by China, Laos and Vietnam" many times in the manuscript. Please simply state that the study was conducted "in Jiangcheng" or "in Yunnan province" as the place where the specimens were collected. If the survey was conducted also in Laos and Vietnam, it would be possible to mention "the border area shared by China, Laos and Vietnam," but the survey was conducted only in China. There is no evidence that EHDV-10 is present in Laos or Vietnam. Please revise the description including other parts of the manuscript.
Introduction
Line 57
The abbreviation for the World Organization for Animal Health should be written as (WOAH) or (WOAH/OIE).
Materials and Methods
Line 92
Culicoides is not mosquito. Please correct the description.
Line 110
Please explain what BHK-21 and C6/36 cells are derived from, respectively.
Lines 123-124
Please correct "Paya serogroup virus" to "Palyam serogroup viruses". Also, this RT-PCR assay can detect Simbu serogroup viruses, but not Akabane virus (Simbu group-specific detection but NOT AKAV-specific detection). Both corrections should be made, as well as other parts of this manuscript.
Lines 131-132
The authors should specify equipment and reagents for the sequence analysis, not where it was performed. Please revise.
Lines 139-140
It is sufficient to write once the location of the same company.
Line 148
Please specify the name of the gel recovery kit.
Line 173
Please state the purpose of the PCR in the title of the section, e.g., "DNA extraction and PCR for identification of Culicoides species."
Lines 174-182
Please specify whether morphological observations were made to identify Culicoides species.
Lines 187-189
Please specify which software you used to calculate the homology, or you just did it with BLAST search. And I think "Table S1.2" should be "Table 3". Please check and correct.
Lines 198-200
Please specify the animal species from which the serum was collected (cattle and goats?) and the number of samples by animal species in this section.
Lines 204-238
The neutralization test has been described in many other papers. Please cite a paper (or papers) and describe the method briefly.
Results
Lines 239-417
In the Results section, there seem to be several sentences that should be included in the Discussion section; in sections 3.3, 3.4, 3.7.1, 3.7.2, and 3.8, the last sentence should be transferred to the Discussion section.
Lines 260-261
The last sentence of this paragraph is unnecessary. When arbovirus is isolated with cultivated cells, CPE is often seen after blind passage(s).
Lines 262-285
This paragraph, Figure 1 and Table 1 are unnecessary. It might be worth stating if the properties were clearly different from other EHDV strains, but I don't think there are any obvious differences.
Line 307
"EHDV strain ON-4" is not the correct strain name. Please correct it, as well as other parts of this manuscript.
Lines 334-344, Table 3
Please specify the name of each strain of EHDV serotypes 1, 2, 4, 5, 6, 7 and 8 used for the calculation of the homology in this study.
Lines 339-341
Complete sequences of Seg-1, 4, 5, 7, 8, 9, 10 of EHDV-10 are available in GenBank (LC552748- LC552754), so please calculate the homology.
Discussion
Lines 515-528
Since there is no information related to the onset of the disease, I do not think the fifth paragraph of the Discussion section is necessary.
Lines 545-618
In the seventh and subsequent paragraphs of the Discussion section and the Conclusion section, the word "sheep" is incorrectly used where it should be "goats". Please correct. Also, there is a statement that the data indicates that cattle, sheep, and “other animals” are infected, but please delete the statement of “other animals”. No evidence is provided at all.

The entire manuscript needs to undergo English proof reading after corrections have been made.
Author Response
Respond to reviewer 1
General comments
Epizootic hemorrhagic disease (EHD) is a WOAH-listed disease and can cause significant economic losses in ruminant livestock production. This study by He et al. is the first report of isolation of a provisional serotype of EHD virus (EHDV) serotype 10 (EHDV-10) outside Japan. This study provides new insights into the distribution of EHDV, but the manuscript contains many rudimentary errors that may confuse readers and should be corrected.
Answer: According to reviewer’s comments, many rudimentary errors that may confuse readers in the manuscript have been corrected.
Specific comments
Abstract
Lines 17-21
The second, third, and fourth sentences are unnecessary; they may be important if the timing of CPE appearance, plaque sizes, and viral titer are clearly different from that of other EHDV strains. Otherwise, they are unlikely to need to be stated.
Answer:It has been revised according to reviewer’s comments in the new revised manuscript. JC13C644 and JC13C673 viruses can cause cytopathic effect (CPE) in mammalian cells BHK21 and Vero cells, and cause morbidity and mortality in suckling mice 48 hours after intracerebral inoculation.
Line 28
Are there any other examples of the expression “Oriental genotype” used for EHDV? I do not think this expression is appropriate. Please correct it, as well as other parts of the manuscript
Answer:According to reviewer’s comments, We have modified ' Oriental genotype ' to ' Eastern group '.
Line 34
The first sentence of the Abstract indicates that Jiangcheng borders Laos and Vietnam. It is acceptable to mention once that Jiangcheng borders Laos and Vietnam as information about the location where the research was conducted, but it is very unnatural to mention "the border area shared by China, Laos and Vietnam" many times in the manuscript. Please simply state that the study was conducted "in Jiangcheng" or "in Yunnan province" as the place where the specimens were collected. If the survey was conducted also in Laos and Vietnam, it would be possible to mention "the border area shared by China, Laos and Vietnam," but the survey was conducted only in China. There is no evidence that EHDV-10 is present in Laos or Vietnam. Please revise the description including other parts of the manuscript.
Answer:It has been revised according to reviewer’s comments in the new revised manuscript.
Introduction
Line 57
The abbreviation for the World Organization for Animal Health should be written as (WOAH) or (WOAH/OIE).
Answer:It has been revised according to reviewer’s comments in the new revised manuscript.
Materials and Methods
Line 92
Culicoides is not mosquito. Please correct the description.
Answer:It has been revised according to reviewer’s comments in the new revised manuscript.
Line 110
Please explain what BHK-21 and C6/36 cells are derived from, respectively.
Answer:It has been revised according to reviewer’s comments in the new revised manuscript.
Lines 123-124
Please correct "Paya serogroup virus" to "Palyam serogroup viruses". Also, this RT-PCR assay can detect Simbu serogroup viruses, but not Akabane virus (Simbu group-specific detection but NOT AKAV-specific detection). Both corrections should be made, as well as other parts of this manuscript.
Answer:It has been revised according to reviewer’s comments in the new revised manuscript.
Lines 131-132
The authors should specify equipment and reagents for the sequence analysis, not where it was performed. Please revise.
Answer:It has been revised according to reviewer’s comments in the new revised manuscript.
Lines 139-140
It is sufficient to write once the location of the same company.
Answer:It has been revised according to reviewer’s comments in the new revised manuscript.
Line 148
Please specify the name of the gel recovery kit.
Answer:It has been revised according to reviewer’s comments in the new revised manuscript.
Line 173
Please state the purpose of the PCR in the title of the section, e.g., "DNA extraction and PCR for identification of Culicoides species."
Answer:It has been revised according to reviewer’s comments in the new revised manuscript.
Lines 174-182
Please specify whether morphological observations were made to identify Culicoides species.
Answer:Due to the small size of Culicoides, many species are similar in the morphology of Culicoides. It is difficult to accurately identify some species of Culicoides only according to the external characteristics of Culicoides without systematic identification such as tableting. Therefore, in this study, we first carried out morphological preliminary identification, and then used molecular biology to accurately identify. The method of preliminary morphological identification was supplemented in ' 2.1.Collection of Culicoides specimens '.
Lines 187-189
Please specify which software you used to calculate the homology, or you just did it with BLAST search. And I think "Table S1.2" should be "Table 3". Please check and correct.
Answer:It has been revised according to reviewer’s comments in the new revised manuscript.
Lines 198-200
Please specify the animal species from which the serum was collected (cattle and goats?) and the number of samples by animal species in this section.
Answer:According to reviewer’s comments, we have specified the animal species from which the serum was collected (cattle and goats) in the new manuscript. We have described the number of samples by animal species in the results section.
Lines 204-238
The neutralization test has been described in many other papers. Please cite a paper (or papers) and describe the method briefly.
Answer:It has been revised according to reviewer’s comments in the new revised manuscript.
Results
Lines 239-417
In the Results section, there seem to be several sentences that should be included in the Discussion section; in sections 3.3, 3.4, 3.7.1, 3.7.2, and 3.8, the last sentence should be transferred to the Discussion section.
Answer:It has been revised according to reviewer’s comments in the new revised manuscript.
Lines 260-261
The last sentence of this paragraph is unnecessary. When arbovirus is isolated with cultivated cells, CPE is often seen after blind passage(s).
Answer:According to reviewer’s comments, the last sentence of this paragraph was deleted.
Lines 262-285
This paragraph, Figure 1 and Table 1 are unnecessary. It might be worth stating if the properties were clearly different from other EHDV strains, but I don't think there are any obvious differences.
Answer: According to the reviewer’s opinion, we delete the Figure 1. Although the reviewer suggested that there was no significant difference in biological characteristics between the new isolated virus strain and other isolates, it was recommended to be deleted. However, Table 1 describes the biological characteristics of the newly isolated EHDV10 strain in Yunnan, and provides readers with relevant information on the biological characteristics of the virus. Therefore, after careful consideration, it is necessary to retain Table 1.
Line 307
"EHDV strain ON-4" is not the correct strain name. Please correct it, as well as other parts of this manuscript.
Answer:According to the reviewer’s comments, we have corrected the strain name.
Lines 334-344, Table 3
Please specify the name of each strain of EHDV serotypes 1, 2, 4, 5, 6, 7 and 8 used for the calculation of the homology in this study.
Answer:According to the reviewer’s comments, we have specified the name of each strain of EHDV serotypes 1, 2, 4, 5, 6, 7 and 8 for homology analysis in this study, and put them below Table 3.
Lines 339-341
Complete sequences of Seg-1, 4, 5, 7, 8, 9, 10 of EHDV-10 are available in GenBank (LC552748- LC552754), so please calculate the homology.
Answer:Complete sequences of Seg-1, 4, 5, 7, 8, 9, 10 of EHDV-10 were obtained from GenBank. These sequences were added and the homology was recalculated.
Discussion
Lines 515-528
Since there is no information related to the onset of the disease, I do not think the fifth paragraph of the Discussion section is necessary.
Answer:According to the reviewer’s comments, we have revised them in the new manuscript.
Lines 545-618
In the seventh and subsequent paragraphs of the Discussion section and the Conclusion section, the word "sheep" is incorrectly used where it should be "goats". Please correct. Also, there is a statement that the data indicates that cattle, sheep, and “other animals” are infected, but please delete the statement of “other animals”. No evidence is provided at all.
Answer: According to the reviewer’s comments, we have revised them in the new manuscript.
Comments on the Quality of English Language
The entire manuscript needs to undergo English proof reading after corrections have been made.
Answer: The manuscript have been reviewed by an native speaker, and many parts of the manuscript have been rewritten to improve the quality of the writing.

Reviewer 2 Report
Comments and Suggestions for Authors
Yuwen He and colleagues present an interesting manuscript on the detection as well as molecular and virological characterization of two virus strains identified as Epizootic Hemorrhagic Disease Virus Serotype 10 which have been isolated from Culicoides tainanus in China. The authors additionally include a serological survey in cattle and sheep (might bei goats) in the same geographical region revealing that antibodies against EHDV-10 is commonly present in these domestic animals. The paper is well written, sometimes almost too detailed, and the conclusions are sound.
There are only some minor remarks:
Sheep versus goats: It is not clear whether the investigations included sheep or goat sera. Please check and use the correct domestic animal name in the whole manuscript and the figure.
Culicoides species: please add some information how the species listed in lines 244-248 were determined. If necessary, please add a corresponding reference for the taxonomic key or whatever was used. From Figure 2 it is not clear why the Culicoides containing the two virus strains were affiliated to C. tainanus and not to C. maculatus. Can you comment on this? The genus name Culicoides should be written in italcis in the whole manuscript
Line 66: Jordan is an Asian country.
Line 101: “Midges displaying analogous morphological characteristics”. Please replace “analogous” by “similar”
Line 292-293: I did not find the GenBank accession number of the COI gene sequences of Culicoides JC13C673 and JC13C644. Could you please add them?
Line 435-436: “In recent years, EHDV-10 virus has been recurrently isolated from surveillance animals in Shizong and Jiangcheng, located in Yunnan Province, China”. Could you please add a reference for this information?
Comments on the Quality of English LanguageThe quality of English language is fine. Minor editing of English might be necessary in some sentences.
Author Response
Respond to reviewer 2
Sheep versus goats: It is not clear whether the investigations included sheep or goat sera. Please check and use the correct domestic animal name in the whole manuscript and the figure.
Answer:It has been revised according to reviewer’s comments in the new revised manuscript.
Culicoides species: please add some information how the species listed in lines 244-248 were determined. If necessary, please add a corresponding reference for the taxonomic key or whatever was used. From Figure 2 it is not clear why the Culicoides containing the two virus strains were affiliated to C. tainanus and not to C. maculatus. Can you comment on this? The genus name Culicoides should be written in italcis in the whole manuscript
Answer:According to reviewer’s comments, we have added a description of the identification of Culicoides in the manuscript“They were preliminarily classified and identified on ice according to their morphological characteristics, such as wing pattern, body size, color and so on.” and added a corresponding reference for the taxonomic key. C.tainanus and C.maculatus are homonymous Culicoides with different name.In order to keep consistent with the downloaded sequence names in the Gene bank, we use the different names of C.tainanus in Figure 2.
Line 66: Jordan is an Asian country.
Answer:We corrected it according to the reviewer’s comment.
Line 101: “Midges displaying analogous morphological characteristics”. Please replace “analogous” by “similar”
Answer:We have replaced it according to the reviewer’s comment in the new manuscript.
Line 292-293: I did not find the GenBank accession number of the COI gene sequences of Culicoides JC13C673 and JC13C644. Could you please add them?
Answer:We have submitted COI gene sequences of Culicoides JC13C673 and JC13C644 to GenBank. Once the GenBank accession number was obtained, we immediately added GenBank accession to the manuscript.
Line 435-436: “In recent years, EHDV-10 virus has been recurrently isolated from surveillance animals in Shizong and Jiangcheng, located in Yunnan Province, China”. Could you please add a reference for this information?
Answer:According to reviewer comments, we have added the references to the new manuscript.

Round 2
Reviewer 1 Report
Comments and Suggestions for Authors
The manuscript has been revised appropriately. I would like to make a minor correction in Line 200. The software is spelled incorrectly, please correct the spelling to "MEGA".
Author Response
The manuscript has been revised appropriately. I would like to make a minor correction in Line 200. The software is spelled incorrectly, please correct the spelling to "MEGA".
Answer: According to reviewer’s comments, We have corrected the software name misspelling ' MAGE ' to ' MEGA '.